# Healthcare-Seeking Behaviors of Homeless Substance Users During the COVID-19 Lockdowns in Gauteng, South Africa: A COREQ-Based Report

Mayibongwe Mkhaliphi Abel Mnkandla, Takalani Grace Tshitangano and Azwinndini Gladys Mudau *

Department of Public Health, Faculty of Health Sciences, University of Venda, Private Bag, X5050, Thohoyandou 0950, Limpopo, South Africa; khabos05@gmail.com (M.M.A.M.); takalani.tshitangano@univen.ac.za (T.G.T.)
* Correspondence: azwinndini.mudau@univen.ac.za

**Abstract:** Substance use continues to be a social problem globally. Around the world, approximately 275 million people use drugs, including 15% of South Africans, with over 36 million succumbing to drug disorders. In the Gauteng Province, about 55% of autopsies of deceased medicolegal cases (2003–2012) showed the presence of one or more illicit drugs in Pretoria. Most research shows that around one-third of people who have problems with alcohol and drugs are homeless. Evidence suggests that homeless people are often denied access to even the most essential assistance, including healthcare. This study assessed the healthcare-seeking behaviors of homeless substance users during the COVID-19 lockdowns in Gauteng, South Africa. A qualitative approach using an exploratory design assessed referrals based on those suffering from symptoms of COVID-19 during the lockdown period between 2020 and 2022. Data were collected from 25 homeless substance users in the City of Tshwane, Gauteng, through unstructured interviews. Three themes based on the study objectives included the types of healthcare services consulted, the determinants of health-seeking behaviors, and challenges experienced while seeking healthcare. Of the twenty-five participants, aged twenty-one to fifty, thirteen suffered from COVID-19 symptoms. Five used formal healthcare systems and eight used informal healthcare systems. Older participants access healthcare services, while less of the younger population use them. Barriers that are encountered while seeking medical services entail marginalization, stigmatization, and a lack of social support. Despite the formidable barriers posed by homelessness and substance use, homeless individuals demonstrated remarkable resilience in their efforts to access healthcare services during the COVID-19 lockdowns. This study highlights the importance and urgent need for harm reduction strategies and policy development for long-term service provision to this population, as well, as the literature on equity, diversity, and inclusion as a foundation for the rights of marginalized populations and groups. Future research directions should focus on harm reduction among this population group of homeless substance users.

**Keywords:** COVID-19; Gauteng; healthcare-seeking behaviors; homeless; South Africa; substance users



## 1. Introduction

Health-seeking behavior is any action undertaken by individuals who perceive themselves to have a health problem or to be ill to find an appropriate remedy (Haileamlak 2018). Good health-seeking behavior is essential for the prevention, early diagnosis, and management of disease conditions. It helps reduce costs, disability, and death from diseases (Haileamlak 2018). Regarding substance users, stereotypes suggest poor general healthcare-seeking behaviors about tuberculosis and HIV/AIDS. COVID-19 is a novel communicable disease, and researchers also need to document health-seeking behaviors in different population groups.

Homelessness continues to be a global social security problem affecting both developed and developing countries (Tenai and Mbewu 2020). There were 274,000 homeless people in England in December 2021 and 365,535 in 2023 (World-Population-Review 2023a). In America, 582,462 individuals are experiencing homelessness in 2023 (World-Population-Review 2023a). Nigeria has the highest number (24,400,000) of homeless people in the world, followed by Pakistan (20,000,000), Egypt (12,000,000), and Syria (6,568,000) (Howells et al. 2023). Grenada has the lowest number of homeless people (68), followed by Ivory Coast (117). In South Africa, (Tenai and Mbewu 2020) noted that people of all ages, young and old, including mothers with young children, experience homelessness, while countries such as Jordan, Cuba, Bhutan, and Liechtenstein have been declared to have the fewest homeless people as of 2023 (World-Population-Review 2023b).

The Human Sciences Research Council (HSRC) estimates that between 100,000 and 200,000 people live on the streets of South Africa (Rule-Groenewald et al. 2015). Accurate statistics are non-existent, though (Mitchely 2021) states that there may be 50,000 people living on the streets of Gauteng. (Shoba 2021) estimates that Johannesburg has 15,000 people in the streets, with Tshwane having 10,000. According to (Shoba 2021), NGOs indicate that 14,000 people in Cape Town are homeless, the number growing exponentially since the start of the COVID-19 pandemic due to job losses and other knock-on effects of the virus.

Homeless people quickly access drugs, with abuse being a survival measure to escape the reality of being homeless and issues arising (Tenai and Mbewu 2020). (Makiwane et al. 2010) postulate that robbery and drug trafficking are the most dominant survival systems for people experiencing homelessness. Substance use cases have risen globally, with people falling prey to their addiction. Around the world, approximately 275 million people use drugs, with over 36 million succumbing to drug disorders, while in South Africa, 15% of the country's population is associated with drug use (United-Nations 2021). In Gauteng, in the Atteridgeville area, (Moodley et al. 2012) noted the lifetime prevalence rates for substances to be 51.4% (95% confidence interval (CI) 41.5–61.5%) for alcohol, 25.2% (95% CI 17.1–33.3%) for cigarettes, and 13.2% (95% CI 8.3–18.2%) for cannabis among secondary school learners. Alcohol has the lowest mean age of initiation at 14.6 years (standard deviation 2.0). About 55% of deceased medicolegal cases (2003–2012) autopsies showed the presence of one or more illicit drugs in Pretoria, with demographics presenting 51.9% cases being males between 20 and 30 years of age who are amongst the economically active groups (Liebenberg et al. 2016).

In the Gauteng Province of South Africa, COVID-19 cases developed quite notably in the Tshwane district before the third wave in 2021, approaching the 20,009 mark and having more critical patients hospitalized (NICD 2021). Due to the prevalence of the virus in wintry conditions, homeless people are notably vulnerable or more susceptible as the second wave spreads throughout the cold season. Therefore, the Government of South Africa instituted a National lockdown on 27 March 2020, while under the Disaster Management Regulation Section 11 (d) mandated the relocation of homeless people to temporary shelters, regulated by the Department of Social Development. Organizations such as TB/HIV Care and the Community Oriented Substance Use Programme (COSUP), which were already working with substance users on harm reduction in diseases such as HIV through the needle and syringe programme, quickly mobilized resources to facilitate the relocation of the homeless to Caledonian Stadium (Elizabeth and Open Society Foundations 2020). A partnership with the City of Tshwane Metropolitan Municipality and the Gauteng Department of Health's Tshwane District in South Africa was formed to cater to substance users. However, the exercise of identifying and moving homeless people became repetitive since many were absconding from the temporary shelters (Thompson 2020). Some were leaving shelters because of substance abuse and subsequent cravings, and others preferred to stay in their own small community. According to (Mitchley 2020) temporary shelters faced the challenge of substance abuse, with some homeless people suffering from withdrawal symptoms, which at times manifested in aggressive behaviors and violence.

Access to healthcare and social services is one of the five immediate demands in the homeless manifesto launched by the National Homeless Network in South Africa (Shoba 2021). Of particular concern is that people experiencing homelessness are vulnerable to communicable respiratory diseases (such as COVID-19) and have low chances of accessing healthcare services (Mitchley 2020). Therefore, health rights stipulated in the South African Constitution are blurred for people experiencing homelessness as some do not possess identity documents, making it hard for them to attend healthcare facilities (Khoza 2016). Postponing health as a priority is worrying in relation to communicable diseases (such as COVID-19), as an increase in infection is prone to happen upon late access to healthcare attention, resulting in prospective mutations and intensifying the pandemic (Prentice and Pizer 2017). Global statistics report that substance users' mortality rate is fueled by the wide gap between the onset of disease symptoms and seeking relevant, adequate medical attention (Connery et al. 2020). Globally, the number of substance users succumbing to diseases has risen to 15%, from tuberculosis to HIV/AIDS and seasonal flu (Badane et al. 2018). Various socioeconomic factors play a pivotal role in influencing such behavior patterns. In India, (Dubey et al. 2020) noted that factors contributing to poor healthcare-seeking behaviors are linked to populations that have fallen prey to drug addiction.

### 1.1. Rationale

With discoveries in 2016 noting that 3.6 million deaths in low- and middle-income countries (LMICs) were due to the non-use of healthcare, postponing healthcare as a priority leads to increased mortality among homeless substance users. The study will provide insights into how lockdown measures impacted homeless substance users' ability to access healthcare services, including testing, treatment, and preventive measures related to COVID-19. Homeless individuals and substance users are considered high-risk groups for COVID-19 due to their living conditions and compromised health status. Examining their healthcare-seeking behaviors will shed light on potential gaps in public health strategies and inform future pandemic preparedness efforts. It is key to understand their experiences during the pandemic, which can contribute to more inclusive and effective healthcare policies targeting this population.

This study mainly investigates the health-seeking behaviors of homeless substance users residing in Gauteng Province of South Africa during COVID-19 lockdowns. The objectives addressed are:

- To describe the types of healthcare services consulted by homeless substance users who experienced COVID-19-related symptoms.
- To describe the determinants of health-seeking behaviors amongst homeless substance users.
- To explore challenges faced by homeless substance users while accessing healthcare services.

### 1.2. Significance

The study findings serve as baseline data for non-governmental organizations, the private sector, and government interventions to promote positive health-seeking behavior among substance users living in the informal settlements of the Gauteng Province. Information on healthcare-seeking beliefs can be used in program planning for communicable disease intervention strategies and strengthening existing strategies during NGO monitoring and evaluation.

### 1.3. Theoretical Framework

Andersen's expanded behavioral model of health service use was used as a framework for this study. To predict or explain one's use of healthcare services, the model focuses on an individual's predisposition to use acute healthcare services, enabling factors that facilitate use, and one's perceived or influenced need for care (Andersen 1995; Andersen and Newman 1973). In this study, Anderson's expanded behavioral model of health service use

was used to design the purpose and objectives of this study and the subsequent presentation of findings. Thus, the social determinants describe the predisposing factors, the challenges represent the enabling factors, and the healthcare services consulted represent or explore the need.

## 2. Materials and Methods

### 2.1. Study Approach and Design

The methodology and material are reported following the Consolidated Criteria for Reporting Qualitative Research (COREQ) to improve rigor. A qualitative approach utilizes facilitated data collection from substance users to explore their healthcare-seeking behaviors and contributing factors. The illustration of the depth and breadth of the participants' subjective experiences (Funk and Kobayashi 2016) enabled a deeper understanding of the healthcare-seeking phenomenon, and a range of factors affecting COVID-19 were discovered (Creswell and Poth 2020). Thus, the exploratory design provides a specific account of homeless people's lived experiences, representing their rights ethically (Gray et al. 2016).

### 2.2. Study Setting

The study occurred in Pretoria Central, where most homeless people who use substances are found, with the pre-test in Centurion. Pretoria Central is in the Gauteng Province, which is divided into three metropolitan municipalities: The City of Ekurhuleni, the City of Johannesburg, and the City of Tshwane Metropolitan Municipalities, as well as two district municipalities, which are further subdivided into six local municipalities: Emfuleni, Lesedi, Midvaal, Mogale City, and West Rand City. Figure 1 below shows a Map of Gauteng municipalities.

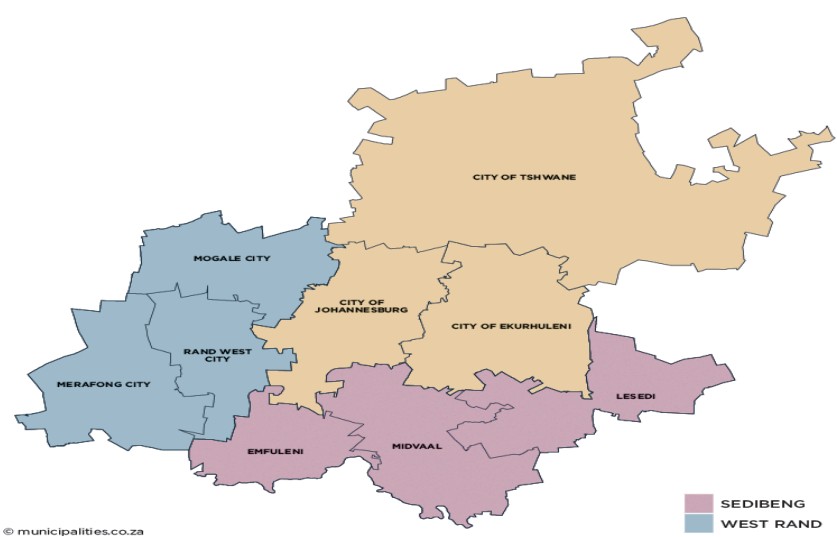

**Figure 1.** Map of Gauteng Municipalities.

Though Gauteng is the smallest of South Africa's provinces, covering an area of 18 178 km$^2$, where there are opportunities for small hand-to-mouth businesses, 13,399,725 people—24.1% of the national population—live in this province. Gauteng is bordered by the Free State, Northwest, Limpopo, and Mpumalanga provinces. There are around 50,000 homeless people in Gauteng, 15,000 in Johannesburg, and 10,000 in Tshwane. Though homeless people are scattered throughout Gauteng Municipalities, they are more concentrated in the city centers or near shopping centers (Tenai and Mbewu 2020). Thus, the study was conducted in Tshwane, Lyttleton, and Sunnyside city centers, where most homeless substance abusers seek subsistence opportunities.

### 2.3. Study Population and Sampling

The study population comprised homeless people between 18 and 64 years of age within the in-city centers of Tshwane during the COVID-19 lockdowns. The target population is homeless substance abusers around Tshwane in Pretoria.

In this study, selective sampling involved snowballing with scrutiny processes, where only those homeless people referred for recruitment were included based on their capability of articulating the facts needed and having suffered from COVID-19 symptoms during COVID-19 lockdowns. As per the guidelines of exploratory design, snowballing continued from the participants selected purposefully up to 23 referrals using the snowballing technique, with referrals going through the same screening questions. Interview questions required yes/no answers on substance use and past infection with COVID-19. Referred substance users were interviewed until data saturation was reached, ending the sample frame.

### 2.4. Data Collection Instrument

An interview guide made up of open-ended, semi-structured questions developed by the researcher initially assessed demographic data such as gender, level of education, and informal place of living. From question 4, the participants were interviewed on substance use and how long they had been using their choice of substance, exposure to COVID-19, and response mechanisms taken. The exposure was assessed through symptoms that triggered health-seeking behavior, types of healthcare consulted, determinants of the chosen type consulted, and the duration between the onset of symptoms and seeking healthcare attention.

### 2.5. Pre-Test

A sample of two randomly selected participants in Centurion, Pretoria, where substance users were offered food daily, was used to test the interview guide. Ambiguous questions were rephrased for clarity. During the pre-test, the following were noted: ease of accessing the facilities, language, time taken during the interview, recording/taking notes, conversational flow, and ambiguities (Hoepfl 1997).

### 2.6. Trustworthiness

The trustworthiness of the study findings was ensured through credibility, transferability, dependability, reliability, and conformability. Credibility was achieved through interaction with participants before the interviews and by allowing those who gave usable data to go through the audio recordings. Transferability was ensured by comprehensively describing the methodology, data collection, data management, and analysis until the report was developed. The researcher revised the analyzed information to assess other angles of data analysis that gave comparable results to ensure dependability. Finally, for conformability, constant revision of the transcripts for quality audits was performed by the principal investigator through listening to recordings repeatedly to ensure that the data collected support the argument backed by literature consulted through scoping review upon discussion of findings and repeatability.

### 2.7. Data Collection

Skilled healthcare workers conducted in-depth individual interviews in places of substance users' habitation. These community healthcare workers under Best Health Solutions (an NGO), holding National Diplomas in Social Work, had been working with homeless substance users on programs such as food distribution since before the COVID-19 lockdowns in South Africa. Thus, an already existing professional relationship was developed to ease the opening-up and transparency of the participants and provide in-depth and true data. Ethical clearance was obtained from the University of Venda (FHS/22/PH/10/2609) and permission to conduct the study was obtained from Best Health Solutions, which was

already working with the participants in these setups, representing the Department of Health in Gauteng for provincial permission.

Prior to conducting the research, each participant was provided with detailed information about the study's purpose, procedures, potential risks, and benefits. They were given sufficient opportunity to ask questions and clarify any doubts they had. Only after understanding the study's objectives and procedures, and freely agreeing to participate, did the participants provide their written consent. This process aimed to maintain the participants' rights and autonomy throughout the research process. Data was collected in the streets and abandoned buildings over two weeks in January 2023 after Best Health Solutions was consulted to facilitate the process. Demographic data were collected first from participants without names and other confidential information, followed by the main question, "*Did you suffer from flu-like symptoms during COVID-19 pandemic lockdowns?*". A positive response allowed participants to be probed to give more information on their choice of healthcare-seeking behaviors and input on easing the process. Most participants understood isiZulu as obtained from the pre-test on two participants. Thus, an exchange with English training was conducted with community health workers to obtain uniformity according to the preference of participants and address probing. Data from the pre-test were included in the primary analysis to ensure valid information was included in the study. Participants uncomfortable with voice recordings and probing questions could withdraw their consent. Interview questions required yes/no answers on past infections or symptoms stated. Referred substance users were interviewed until data saturation was reached, ending the sample frame. Interviews were held for a maximum of 18 min until data saturation was attained; the target was 20 random sets without any confidential information being questioned. Participants were thanked for their cooperation at the end of each session.

*2.8. Data Management and Analysis*

All voice recordings were translated, transcribed, and matched to the interview transcripts developed. Participants who gave usable data could go through the recordings to validate the data from the questions. All recordings are stored for five years in a Google Drive account logged into by the student's Mvula account, sharing access with the supervisors, library, and relevant stakeholders. Data analysis ensures grouping, synthesizing, and filling data collection materials in a well-planned manner, facilitating ease of duplicity and retrievability. The data analysis methods employed were interpretative phenomenological data analysis (IPA), which entailed using a phenomenological research design to do a detailed scrutiny or research of homeless substance users' healthcare-seeking experiences. Although (Pietkiewicz and Smith 2014) argue that the IPA design adopts only specific experiences, this study explored a comprehensive approach through multiple readings and taking notes, interpreting notes into themes, seeking connections, and clustering themes to write the IPA results. Familiarization with the recordings was performed solely by the principal researcher, and audio was repeatedly played while taking notes. Translation from isiZulu to English and transcription were performed intelligently and verbatim. Relationships across themes were performed, with transcripts further reviewed by participants to confirm accurate data extraction. All the codes noted in various themes were grouped and described under themes. Formal and informal healthcare services were grouped under the theme "types of healthcare services consulted", marginalization, stigmatization, and social support under "challenges faced while accessing healthcare services", and the level of education vs. age was grouped under "social determinants of health", which are the socioeconomic conditions influencing differences in individual and group health access and status such as age group, level of education, homelessness, and access to resources or wealth. No analysis software or contractors were used.

## 3. Results

### 3.1. Characteristics of Participants

Table 1 describes the participants' demographic characteristics as explored by the questions about their age and education. Regarding the predisposing factors, participants are aged between 21 and 40 years, with five having tertiary education, eight having matriculated, and twelve having not reached matric levels of education. The spread of healthcare-seeking behaviors will be explored later across various education levels.

**Table 1.** Substance users' demographic characteristics.

| Participant ID | Age | Level of Education |
| --- | --- | --- |
| 1 | 35 | Matric |
| 2 | 36 | Below matric |
| 3 | 38 | Below matric |
| 4 | 38 | Below matric |
| 5 | 50 | Below matric |
| 6 | 32 | Below matric |
| 7 | 31 | Matric |
| 8 | 27 | Matric |
| 9 | 33 | Tertiary |
| 10 | 25 | Matric |
| 11 | 24 | Tertiary |
| 12 | 31 | Below matric |
| 13 | 27 | Below Matric |
| 14 | 29 | Matric |
| 15 | 24 | Below matric |
| 16 | 21 | Below matric |
| 17 | 38 | Matric |
| 18 | 35 | Tertiary |
| 19 | 27 | Matric |
| 20 | 40 | Matric |
| 21 | 36 | Tertiary |
| 22 | 21 | Below matric |
| 23 | 33 | Below matric |
| 24 | 23 | Below matric |
| 25 | 40 | Tertiary |

### 3.2. The Health-Seeking Behaviors of Homeless Substance Users in Gauteng Province

In this study, Andersen and Newman's seeking behavior model framework was used to explain the objectives (Andersen 1995; Andersen and Newman 1973). This entails that the type of healthcare services consulted represents the need, the determinants represent the predisposing factors, and the challenges represent the enabling factors. Thus, the theoretical framework-based findings of this study may be presented in Figure 2 below, where the need, predisposing factors, and enabling factors are themes and their accompanying sub-themes.

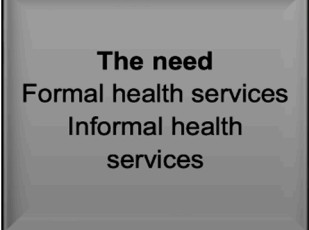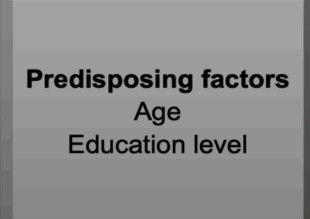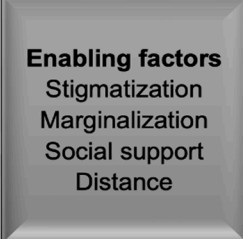

**Figure 2.** Andersen and Newman seek a behavioral model based on themes and sub-themes.

Table 2 below summarizes the findings in terms of objective-based themes.

**Table 2.** Themes and sub-themes originating from data analysis.

| Themes →  | Types of Healthcare Services Utilized | Social Determinants of Healthcare-Seeking Behaviours | Challenges Faced Accessing Healthcare Services |
|---|---|---|---|
| **Sub-themes** → | Formal services | Level of education | Stigmatization |
| | Informal services | Age | Marginalization |
| | | | Empathy and social support |

### 3.2.1. Theme 1: Types of Healthcare Services Utilized

Twenty-five participants were interviewed, with thirteen having suffered from COVID-19 symptoms and accessing various healthcare services; thus, their responses were referred for data analysis. All thirteen referred participants with COVID-19 symptoms used various types of healthcare services. The types were grouped into formal and informal healthcare services. The questions asked include *"Who did you consult to get health care attention?"* and *"What did you do to get better?"* Answers were interpreted into the sub-themes of formal and informal healthcare systems.

*Sub-theme 1: Formal healthcare systems*

Five participants knew and confirmed using these services during the COVID-19 lockdowns. These were noticed as consulting hospitals, clinics, and community healthcare workers, as described below.

*Clinics and Hospitals*

A handful of four out of thirteen referred participants were able to approach hospitals and clinics after feeling symptoms of COVID-19, showing appreciation for formal healthcare services. They proved effective, as the participant (P7), due to vaccination, did not get sick, while P4 never had problems again after being vaccinated against COVID-19.

According to P4, "…I steamed, isn't they instructed us to cover our mouths, I also drank med lemon and went to the hospital where I was given iron tablets and I became better. For the second time, I was given an injection and sets of pills, that is when I got vaccinated…."

Participant (P7) highlighted that visiting the hospital and getting vaccinated led to not getting COVID-19 symptoms. Getting vaccinated was a family decision for every member of the family to get vaccinated, being a comprehensive factor of family influence in seeking healthcare aid from hospitals. According to (P7),

*"…I went to the hospital. It wasn't my choice; it was the household (family decision) choice bra. It's either you do it or you going to go …."*

*"…Not at all, because I went to vaccination…."*

One participant (P21) was already accessing healthcare services for other comorbidities, thus being alerted to preventative measures against COVID-19. The strategies taken from



thenceforth were self-isolation as the norm and living in abandoned shelters. When asked which services were sook and if they had suffered from COVID-19, respectively:

(P21) "...*the fortunate part is I got injured, so I had to go to the hospital, so that was the only way I could confirm that I am not contaminated....*"

(P21) "...*Unfortunately, not because we are junkies, we like to privatize ourselves like we were being indoors....*"

Moreover, preventative measures learned from hospitals were mentioned:

(P21): "... *Mostly it was to isolate me, stay somewhere where there are no people and try to get warm be away from people because you did not know how you can get this. So, it was me like trying to protect myself because that was the message around that time....*"

Participant (P15) responded: "*I went to the hospital and consulted a doctor.*"

*Community healthcare workers*

Community healthcare workers served as a means of healthcare distribution during the national lockdown period. Holistic measures had to be taken to spread disease prevention and treatment awareness. Thus, community healthcare workers also educated homeless substance users on such strategies. According to participants (P4) and (P18), respectively, when asked, "*Whom did you consult to get health care attention?*",

"...*I steamed, isn't they instructed us to cover our mouths...*"

"...*I did see this sister who gives us food and made me drink things like medication...*"

*Sub-theme 2: Informal systems*

Informal systems influence self-diagnosis and result in self-prescribing either over-the-counter or traditional medicine, other informal medicines, or nothing. They are mainly influenced by the social characteristics of population groups, such as cultural and self-beliefs. In this study, this is clear from participants who did not mention using clinics/hospitals and consulting community healthcare workers when asked, "*Whom did you consult to get healthcare attention?*" and "*What did you do to get better*?"

*Self-care, family, and friends*

Eight substance users relied on the advice they shared while some were dealing with the pandemic. Decision-making is influenced in circles where people share ideas as a community. The assessment questions that brought up behavioral aspects under the influence of community setup are:

"*So, who did you consult to get health care attention "and "What did you do to get better from the symptoms?*".

*Self-prescribed medicine*

When assessed on strategies taken and treatment methods, common self-prescribed means of treating symptoms included taking med lemon (hot medication which relieves flu symptoms), and cough syrup.

In the study, participants who made use of self-prescribed remedies are (P6, P9, P23, and P24), with their respective responses from the assessment being:

(P6): "...*I did not go to the clinic .... Just the way you treat flu, I took Med Lemon....*"

(P9): "...*no one... I drank med lemon...*"

(P23): "... *I did not go to the clinic; I find means to help myself... I take Med lemon or take drugs....*"

(P24): "...*no one... I got myself a cough mixture...*"

*Physical distancing*

Physical distancing involves strategies such as isolation and quarantining. This strategy was practiced by one participant (P21), who had other comorbidities except for COVID-19. However, before seeking healthcare attention, the participant (P21) isolated themselves to reduce the risk of transmission and burden from the symptoms they experienced. The response given when assessed on the strategy used was:

*". . . Mostly, it was to isolate me, stay somewhere where there are no people, and try to get warm be away from people because you did not know how you could get this. So, I was trying to protect myself because that was the message around that time. . .."*

*Unknown substances*

In the study, two participants (P2 and P20) relied on self-diagnosis, leading them to choose any closest means available to treat themselves by reference, such as unknown substances, as the active ingredients were unknown. When asked what strategies they used or whom they consulted:

(P2): *". . .there is no one I spoke to, sometimes you see I get pills from this other man whom he uses and gets better . . ."*

(P20): *". . .me, myself, and I. . . . . .. I cleared my chest for hay fever because it was very dark. . .."*

Two out of thirteen referred participants treated themselves using unknown substances. The effectiveness of these substances against COVID-19 is not known.

### 3.2.2. Theme 2: Determinants of Health-Seeking Behavior amongst Homeless Substance Users

As all the participants were homeless as per the inclusion criteria of the target population, accommodation was uniform across them. Therefore, social determinants of health were assessed by the questions *"Please confirm your age, how old are you*?" and *"What is your level of education?"* Answers on level of education and age were then linked to the type of healthcare services used. Table 3 summarizes the spread of the social determinants of health guiding this study.

**Table 3.** Determinants of health-seeking behaviors.

| Participant ID | Age | Level of Education | Type of Healthcare Service Used | |
| --- | --- | --- | --- | --- |
| | | | Formal | Informal |
| 22 | 23 | Below matric | | Self-prescribed |
| 24 | 23 | Below matric | | Self-prescribed |
| 15 | 24 | Below matric | Hospital | |
| 14 | 29 | Matric | | Unknown |
| 7 | 31 | Matric | Hospital | |
| 6 | 32 | Below matric | | Self-prescribed |
| 9 | 33 | Tertiary | | Self-prescribed |
| 23 | 33 | Below matric | | Self-prescribed |
| 18 | 35 | Tertiary | Community healthcare workers | |
| 2 | 36 | Below matric | | Self-prescribed |
| 21 | 36 | Tertiary | Hospital | |
| 4 | 38 | Below matric | Community healthcare workers and hospital | |
| 20 | 40 | Matric | | Self-prescribed |

The age range was divided into three points: The lower quartile, median, and upper quartile values of 21, 30, and 40. One referred participant below 30 (P15) used formal

healthcare services, and three (P14, P22, and P24) used informal services. Of which two were used within the upper quartile range, four participants (P4, P7, P18, and P21) used formal healthcare services, while five (P6, P9, P23, P2, and P20) used informal healthcare services.

Of the seven participants without matric, five (P2, P6, P22, P23, and P24) used informal healthcare services, while two (P4 and P15) used formal healthcare services. Among the matriculated participants, two (P14 and P20) used informal healthcare services, while one (P7) used formal healthcare services. Among the three participants who had attended tertiary education, two (P18 and P21) used formal healthcare services, while (P9) used informal healthcare services.

### 3.2.3. Theme 3: Challenges Faced While Accessing Formal Healthcare Systems

A level of resistance to perceived strategies is always met wherever there are mitigation systems for a particular matter, even in public health. It can be either reactive or a form of proactive choice due to different belief systems. Substance users noted various relevant challenges while trying to access formal healthcare systems during the COVID-19 lockdowns. The data in this section are based on the question, "*What challenges did you face in accessing health services in general*"?

Some of the challenges faced were a barrier to formal healthcare aid, thus causing substance users to forgo them and use informal healthcare systems during the COVID-19 lockdowns. Challenges met entail procrastination and a lack of efficiency by healthcare professionals, as presented by participant (P23), who mentioned that:

> "…*Some of them don't give use attention and some of them marginalize and keep telling us of the side effects of the drugs we take….*"

Other participants interpreted such measures as stigmatization, as recorded by the participant (P20), who mentioned that:

> "…*But even in the clinic there's just some stigma in trying to find out what's wrong with you, but a lot of people are too scared to go…*"

Other matters arising while accessing formal healthcare systems entailed empathy for this key population group. Other participants mentioned discomfort with communication, which reduces their reliance on formal healthcare systems. One participant said:

> (P4): "…*they were instructing me to wear masks, treating me like a rotten person, but at the end of the day, they helped me….*"

Some participants do not appreciate being checked for other comorbidities. Although this became an integrated approach to healthcare for prospective patients, other patients did not like it. In this study, participant (P15) voiced that:

> "…*entering the hospital gate, the guard discovered that I had high blood and other symptoms. However, they treated me right….*"

According to the participant, physical distancing was enforced in most institutions, resulting in long queues due to gaps between two individuals or more (P21).

> "…*eish it was queues, a lot of queues were very different besides…*"

### 3.2.4. Theme 4: Strategies to Increase Access to Healthcare Services

Insights into ways of reducing challenges in accessing healthcare services were explored. Various concepts spawned as assessments were made to have an autopsy of what substance users thought would help mitigate disease burden upon access to healthcare services. Already existing strategies to reach key populations have been using mobile community healthcare workers and social media for those who can afford smartphones. The question, "*What can be done to increase access to health services?*" was asked, noticing diverse answers. According to various participants, it was noted that:

Two participants (P4 and P23) suggested using community healthcare workers who are empathetic, passionate, and understand the lives of homeless substance users. Psychosocial training of community healthcare workers is challenging because it involves more content and prepares them to deal with various key populations, such as homeless substance users. These participants said:

(P4) *". . . For them to get empathetic personnel, just not anyone. When working in this department, it requires someone with patience and with good listening skills. Without patience, it is not conducive. . ."*

(P23) *". . . for them to employ empathetic and have time for us, people who will understand us. . .."*

Two participants' (P15 and 21) responses focused more on proactive measures catering to substance users, with the participant (P21) being broad on issues requiring improvement, such as mobile strategies. In this case, improvement touches on programs involving homeless substance users to assess their efficiency level. Besides that, constant monitoring of the healthcare status of these key population groups is brought up. Two participants mentioned that:

(P15) *". . .They should just keep checking on the guys without them standing up for themselves to try and go to the hospital. . .."*

(P21) *". . .. for me it is because you have old people who cannot access or have long queues. So, site visits campaigns could help more, and we need more health workers to be hired, and basically, if they could take more students just to uh pre-train them something like that, just for them to get to know the work that they were doing as much as they are studying being healthcare workers. So, more bursaries, more opportunities regarding health so that you can have more. . .."*

One participant (P22) highlighted issues to increase accessibility for them by bringing out mobile primary healthcare facilities moving around to offer healthcare services to such key population groups through:

(P22) *". . . maybe they should bring us mobile clinics. . ."*

## 4. Discussion

The present study aimed to investigate the healthcare-seeking behaviors of homeless substance users during the COVID-19 lockdowns in Gauteng, South Africa. Additionally, the study explored the various types of healthcare services used by homeless substance users, the social determinants of health driving them to do so, and the considerable barriers they faced while accessing their healthcare services of choice.

The restrictions imposed to curb the spread of the virus, including movement limitations and facility closures, drove this population group into consulting various healthcare services with already exacerbated and limited access to medical services. In Marseille, homeless people have been associated with familiarity with infectious diseases and associated protective measures, translating into good skills and practices in dealing with COVID-19 (Allaria et al. 2021). Participants could approach hospitals and clinics for aid amid a pandemic, even though some had various comorbidities they were facing. Findings by (Dada et al. 2022) on a few substance users seeking healthcare aid are still visible comparing the number of participants using formal facilities and not, with a ratio of 5:8, respectively. Good relations thus continue to be implemented between substance users and formal healthcare structures. In terms of targeting and reaching key populations with related health interventions, according to the South Africa National Strategic Plan, healthcare-seeking behaviors have been impressive, with programs such as COSUP making it easier for substance users to approach sites for healthcare aid (Briginshaw et al. 2021). Although healthcare facilities were used, some participants preferred services from community healthcare workers they had worked with in other programs who dynamically aided in COVID-19 mitigation. This strategy is close to findings by (Stonehouse et al. 2023) on

community-oriented means of overcoming the barriers homeless people face in accessing health and social services. Therefore, discoveries are being made on reducing postponing seeking healthcare services, slowly but surely nullifying concepts on the rise of substance users postponing the healthcare (Arde 2020). Community healthcare workers also serve as a critical source of legal services, supplying credible information on health education and mitigation strategies. (Davis et al. 2021) support their involvement as they increase early interventions in reducing disease burden.

The study realized the following are common formal healthcare strategies: Vaccination, medication, and other medical strategies. Vaccination was among the critical formal healthcare strategies in preventing infection with COVID-19, as participants who got vaccinated did not experience any symptoms, thus not developing symptoms or further consulting hospitals after vaccination. This behavior sheds light on the effectiveness of COVID-19 vaccines against the incidence and mortality rate presented in Iran (Rahmani et al. 2022). When supported with medication targeting symptoms and reducing virus prevalence, as participants did, more effectiveness is noted from those who visited hospitals. However, the pill curing COVID-19, ivermectin, still faces pharmacological contradictions, as supported by (Popp et al. 2021), who encourage further adoption of developing COVID-19 medicine.

Findings reveal that during the COVID-19 lockdowns, public health programs were hybridized to ease pandemic needs. Community healthcare workers in homeless feeding schemes also offered health education, mitigation strategies, and medicine to homeless substance users. (Taufik et al. 2022) also recorded that applying proper cough etiquette through wearing masks and covering their mouths prevents the spread of COVID-19, which was carried out by participants under the advice of community healthcare workers. (Marcus et al. 2020) echo practical communication skills and providing good services as critical to reducing absconding services. This study sees this as a mitigation strategy by the government in partnership with COSUP for substance users who abandoned Caledonian Stadium due to poor sanitary services, a shortage of their drug of choice, and poor initiation into the program. Improvement in such facilities or considering well-furbished facilities also motivates prolonged care.

The use of informal healthcare strategies, primarily based on advice from oneself, family, and friends, was also seen. Some findings support the concept of physical distancing as the future of the spread reduction (Collignon 2021). This strategy was noted in participants who preferred to be isolated from the public as they felt symptoms of COVID-19. Therefore, this is effective if combined with over-the-counter medicines such as med-lemon and cough syrups to reduce disease incidence rates. However, poverty characteristics could inhibit access to such drugs, as revealed in the access to healthcare services (Satre et al. 2021). The study also shows informal healthcare services used by homeless substance users with unknown substances illuminating panic and survival mode experienced in different communities during the pandemic. These were in the form of unknown self-prescribed drugs and pills circulating amongst them, taken without knowing what they were. (Kamazima et al. 2020) discovered some conspiracies of *elegant* or *umhlonyane* used to treat COVID-19. Despite their effectiveness to a certain extent, these methods are not confirmed by regulatory boards such as SAHPRA, posing unknown pharmacodynamic effects. Thus, Dr. Mkhize, under the National Department of Health, also did not endorse them, although the President of Madagascar had seen no problem (Mvumvu 2020). Therefore, according to the framework, there is a need for substance users to access healthcare services when feeling sick, whether they are formal or informal.

Three major social determinants of health have been discovered to influence the healthcare-seeking behaviors of people experiencing homelessness in South Africa (Patra et al. 2021). As the behavioral model postulates various factors facilitating or determining which type of services to use prior to utilizing them, these social determinants of health get to play a huge role. Among these, level of education and age became cornerstone social determinants of health, influencing homeless substance users. In this study, 75% of the

youth below the age of 30 had yet to reach their matric level of education, of which 33% used informal healthcare services. Participants were able to make conclusive decisions on infection or not and which facilities to consult, in line with a study in Lusaka, Zambia, where 73% of homeless young adults, based on their level of education, were associated with low-risk beliefs during COVID-19 lockdowns (Samuyachi et al. 2021).

On the other hand, 45% of adults used formal healthcare services. Among the 45%, 55% of the adults had obtained matriculation and further tertiary education. Compared with the youth below 30, there is high use of formal healthcare services among adults, which corresponds to low levels of education among the youth related to low levels of healthcare-seeking perception and use of formal healthcare services (Farrell et al. 2020; Germishuys et al. 2022).

As much as formal healthcare services were workable strategies, substance users faced barriers when accessing them, addressing the third objective of this study. So, various factors and support structures enable or disable people from having access to healthcare services. Participants voiced notions of inequity, unfairness, discrimination, and stigmatization experienced by healthcare professionals when attending formal healthcare systems. Negative experiences and perceptions of discrimination in healthcare settings can lead to postponement or avoidance of medical care altogether (Swartz et al. 2022). Furthermore, participants worried about honest communication from the healthcare administration, where less empathy demotivated their reliance on proper healthcare. Yet this can be avoided, as showing compassion to substance users increases their healthcare-seeking behaviors when attending formal healthcare services (Dumenco et al. 2019).

This study also explored and addressed increasing accessibility to healthcare services through suggestions from participants based on their current challenges, which could help reduce barriers to accessing formal healthcare services. (Davis et al. 2021) highlight targeted interventions as essential to improving healthcare-seeking behaviors. This notion corresponds with substance users, who see it as essential to train fellow substance users in their community to help in programs targeting them. Therefore, this also relates to changing course content and regulatory boards' laws to be holistic toward substance users, as supported by the need to lobby regulatory bodies for policy change (Briginshaw et al. 2021). A drive for internships and in-service training relating to the healthcare rights of substance users is to be addressed to avoid stigmatizing various groups of substance users. Healthcare providers need to be trained to understand the unique challenges faced by homeless individuals and the role that substance use plays in their lives. Creating a non-stigmatizing and supportive healthcare environment can encourage homeless substance users to seek help when needed (Jack et al. 2018).

Therefore, healthcare-seeking is a continuous phenomenon, with various communicable diseases appearing occasionally. (Rogers et al. 2020) explore specific clinical measures to be discovered and implemented. This study corresponds with this strategy as findings present substance users suggesting mobile primary healthcare services assessing their health on the ground proactively and constantly, significantly aiding children and the elderly. This notion provides a dynamic approach to ensuring healthcare services are constantly available for key population groups, addressing equity in healthcare. Acknowledging that substance use among homeless individuals is a complex issue, challenging policymakers to develop intimate strategies addressing such population groups is essential. Addressing the healthcare needs of homeless individuals requires comprehensive and tailored interventions that consider their unique circumstances and challenges. By implementing targeted policies and support systems, we can work towards improving healthcare access and overall well-being for homeless substance users, especially during times of crisis.

## 5. Limitations

One of the primary limitations of this study is the relatively small and potentially non-representative sample size of homeless substance users surveyed. This may limit the

generalizability of the findings to broader homeless populations. The findings may be specific to the unique context of Gauteng, South Africa, and may not be directly applicable to homeless populations in other regions or countries with different healthcare systems, lockdown measures, sociocultural factors, and gender, as more of the study participants were males. Moreover, participation from the youth, ages 18–30, was quite low, which can be further explored.

The study focused on healthcare-seeking behaviors but did not extensively investigate the availability, accessibility, or quality of healthcare services specifically tailored to homeless individuals during the lockdowns. The COVID-19 pandemic and associated lockdowns introduced unprecedented disruptions to daily life. These external factors may have influenced healthcare-seeking behaviors in ways that are challenging to fully separate from the effects of homelessness and substance use.

## 6. Conclusions

In conclusion, this study highlights the importance and urgent need for harm reduction strategies and policy development for long-term service provision to this population, as well as the literature on equity, diversity, and inclusion as a foundation for the rights of marginalized populations and groups. Future research directions should focus on harm reduction among this population group of homeless substance users.

**Author Contributions:** Conceptualization, M.M.A.M., and T.G.T.; methodology, M.M.A.M., T.G.T., and A.G.M.; validation, M.M.A.M., T.G.T., and A.G.M.; formal analysis, M.M.A.M.; investigation, M.M.A.M.; writing, M.M.A.M.; supervision, T.G.T., and A.G.M.; editing, T.G.T., and A.G.M. All authors have read and agreed to the published version of the manuscript.

**Funding:** This research received no external funding.

**Institutional Review Board Statement:** The study was conducted in accordance with the Declaration of Helsinki and approved by the University of Venda Ethics Committee (FHS/22/PH/10/2609) on the 27 September 2022.

**Informed Consent Statement:** Informed consent was obtained from all subjects involved in the study.

**Data Availability Statement:** Data is available upon request.

**Conflicts of Interest:** The authors declare no conflict of interest.

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
