# Peer review of "Healthcare-Seeking Behaviors of Homeless Substance Users During the COVID-19 Lockdowns in Gauteng, South Africa: A COREQ-Based Report"

_socsci, doi:10.3390/socsci12080464_

Round 1

Author Response

All the comments raised were corrected as indicated on the point-by-point response document.

Reviewer 2 Report

Thank you for the opportunity to review this manuscript. Overall, the article provides interesting and important contributions to the literature on the effects of COVID-19 isolation practices on homeless persons in the communities studied. The research is well presented, although there are several areas that require attention. These will be listed below. A strong point of the article is the methodology. It is well articulated and applied to the research. A major area in need of improvement can be the implications for future research and expansion of the effects of the research on future projects and policy development. Harm reduction figures here as it is an important paradigm shift in governance for homeless populations as does the literature on equality diversity, and inclusion.

line(s):

22 reword last sentence to replace "carried out".

42-43 reword to remove ambiguity and contradiction.

83 reword for clarity (habitation?)

89-93 clarify how licit substances fit in this context of illicit drug use.

112 proactive measures suggest harm reduction. A section on this approach is needed here and perhaps in the future research directions.

132 reiterate LMIC so the reader knows what it is.

139 replace semicolon with period.

151 Anderson's model is helpful, but reference to the original work and other authors noted in this reference is needed

244-246 please explain the process of revisions -- by whom, tools used etc...

248-256 this suggests a positive relationship between all participants and the works. Is this true? What about potential bias? Please explain and justify.

284 and other spots -- use another word beside autopsy.

296 SDH requires definition and complete quotation marks.

308 Andersen and Newman cite requires reference.

324 Please clarify exactly how many participants were included in the research. 25 is the sample, but only 13 actually appear in analysis.

463 reword cubbing with another term.

486-487 how can door-to-door campaigns be developed for homeless persons?

513 again, this speaks to harm reduction strategies.

522-524 reword for clarify -- what does ...not other consulting hospitals mean?

575-576, 591 again, harm reduction.

606-610 This section can be expanded to provide more critique about the research.

611-621 A more robust conclusion can include the importance of harm reduction strategies, policy development for long term service provision to this population. As well, the literature on equity, diversity, and inclusion can serve as a foundation for  the rights of marginalized populations and groups.

I wish you well with the revisions and appreciate the the work that has gone into this manuscript.

Many of my comments refer to word choice, syntax, structure and so forth.

Author Response

All comments raised were corrected as indicated in the point-by-point document.

Round 2

Reviewer 1 Report

The authors have sufficiently improved their paper, in response to the comments made.